# COVID-19 and Genetic Variants of Protein Involved in the SARS-CoV-2 Entry into the Host Cells

**DOI:** 10.3390/genes11091010

**Published:** 2020-08-27

**Authors:** Andrea Latini, Emanuele Agolini, Antonio Novelli, Paola Borgiani, Rosalinda Giannini, Paolo Gravina, Andrea Smarrazzo, Mario Dauri, Massimo Andreoni, Paola Rogliani, Sergio Bernardini, Manuela Helmer-Citterich, Michela Biancolella, Giuseppe Novelli

**Affiliations:** 1Department of Biomedicine and Prevention, Tor Vergata University Hospital, 00133 Rome, Italy; latini.andrea@hotmail.com (A.L.); borgiani@uniroma2.it (P.B.); 2Laboratory of Medical Genetics, Bambino Gesù Children’s Hospital, IRCCS, 00165 Rome, Italy; emanuele.agolini@opbg.net (E.A.); antonio.novelli@opbg.net (A.N.); 3Medical Genetics Laboratory, Tor Vergata University Hospital, Rome, 00133 Rome, Italy; rosalinda.giannini21@gmail.com (R.G.); paolo.gravina@ptvonline.it (P.G.); michelabiancolella@gmail.com (M.B.); 4UOC Pediatria, Bambino Gesù Children’s Hospital, IRCCS, 00165 Rome, Italy; andrea.smarrazzo@opbg.net; 5Department of Clinical Sciences and Translational Medicine, Tor Vergata University Hospital, 00133 Rome, Italy; dauri@med.uniroma2.it; 6Department of Systems Medicine, Tor Vergata University Hospital, 00133 Rome, Italy; andreoni@uniroma2.it; 7Infectious Diseases Clinic, Tor Vergata University Hospital, 00133 Rome, Italy; 8Unit of Respiratory Medicine, Department of Experimental Medicine, Tor Vergata University Hospital, 00133 Rome, Italy; rogliani@uniroma2.it; 9Department of Experimental Medicine and Biochemical Sciences, Tor Vergata University Hospital, 00133 Rome, Italy; bernardini@med.uniroma2.it; 10Department of Biology, Tor Vergata University Hospital, 00133 Rome, Italy; citterich@uniroma2.it; 11IRCCS Neuromed, 86077 Pozzilli, Italy; 12Department of Pharmacology, School of Medicine, University of Nevada, Reno, NV 89557, USA

**Keywords:** COVID-19, SARS-CoV-2, genetic variants, host genetic variability, *TMPRSS2*, *PCSK3*

## Abstract

The recent global COVID-19 public health emergency is caused by SARS-CoV-2 infections and can manifest extremely variable clinical symptoms. Host human genetic variability could influence susceptibility and response to infection. It is known that ACE2 acts as a receptor for this pathogen, but the viral entry into the target cell also depends on other proteins. The aim of this study was to investigate the variability of genes coding for these proteins involved in the SARS-CoV-2 entry into the cells. We analyzed 131 COVID-19 patients by exome sequencing and examined the genetic variants of *TMPRSS2, PCSK3, DPP4,* and *BSG* genes. In total we identified seventeen variants. In *PCSK3* gene, we observed a missense variant (c.893G>A) statistically more frequent compared to the EUR GnomAD reference population and a missense mutation (c.1906A>G) not found in the GnomAD database. In *TMPRSS2* gene, we observed a significant difference in the frequency of c.331G>A, c.23G>T, and c.589G>A variant alleles in COVID-19 patients, compared to the corresponding allelic frequency in GnomAD. Genetic variants in these genes could influence the entry of the SARS-CoV-2. These data also support the hypothesis that host genetic variability may contribute to the variability in infection susceptibility and severity.

## 1. Introduction

The ongoing pandemic of coronavirus disease 2019 (COVID-19) certainly represents one of the most important clinical emergencies of the 21st century [1]. The pathogen responsible for this global epidemic is the novel beta-coronavirus, known as SARS-CoV-2. It belongs to the family of Coronaviruses (CoV), a large family of viruses that includes Middle East Respiratory Syndrome (MERS-CoV) and Severe Acute Respiratory Syndrome (SARS-CoV) [2]. COVID-19 disease can manifest extremely variable clinical symptoms, ranging from very mild sub-clinical symptoms to Acute Respiratory Distress Syndrome (ARDS). These highly heterogeneous phenotypes also seem to depend on patient age, gender, and underlying health conditions [3]. Furthermore, this phenotypic variability could be explained by inter-individual genetic unevenness. In particular, host susceptibility and response to infection, could be influenced also by genetic variants in genes coding for proteins with an active role in the entry and spread of SARS-CoV-2.

It is known that SARS-CoV-2 employs its spike type I transmembrane glycoprotein (S protein) to bind the host surface protein ACE2 (Angiotensin-2 Conversion Enzyme) and mediate cell entry [4]. However, recent studies have shown that the entry of SARS-CoV-2 into the target cell also depends on other proteins [5]. Indeed, entry requires S protein priming by cellular proteases, which cleave S protein in S1 and S2 subunits and allows fusion of viral and cellular membranes, mediated by S2. In particular, SARS-CoV-2 uses host serine protease TMPRSS2 for S protein priming and for entry into primary target cells [6].

The high spreading capacity of the infection suggests that there are other mechanisms which actively contribute to viral entry into the cell [5]. In particular, the S protein contains a proprotein convertase (PPC) motif, a functional furin-cleavage sequence (RRAR) located at the junction between the S1 and S2 subunits [7]. Furin (PCSK3) is an ubiquitous membrane-bound protease highly expressed in the lungs. High levels of furin protease in the human respiratory tract may activate the SARS-CoV-2 S protein, cleaving it into S1 and S2 subunits. Hence, furin pre-activation could allow SARS-CoV-2 to be less dependent on target cells, enhancing its entry into target cells with relatively low expression of TMPRSS2, in a mechanism similar to that observed in avian influenza H5N1 virus [8]. 

Molecular bioinformatic studies have identified other possible host-receptor interactions in the entry of SARS-CoV-2 that facilitate the efficiency of viral particles spread egress. These include the dipeptidyl peptidase 4 (DPP4) protein or CD26, used as cell entry receptor by the MERS-CoV [9], and CD147, also known as Basigin (BSG), a transmembrane glycoprotein belonging to the immunoglobulin superfamily which acts as receptor of host cells for SARS-CoV-2 invasion [10,11]. 

In a previous paper we investigated the genetic variants of *ACE2* gene [12] in a cohort of 99 Italian SARS-CoV-2-positive patients by direct exome sequencing, and we found no relevant association between the observed variants and COVID-19 disease [13]. The aim of the present study was to analyze, in an Italian cohort of 131 patients (99 studied in the previous paper for ACE2 and 32 newly recruited), the coding region and the exon/intron junction of the other four genes (*TMPRSS2, PCSK3, DPP4, BSG*) [12] involved in the SARS-CoV-2 entry into target cells.

## 2. Materials and Methods

### 2.1. COVID-19 Patients

We enrolled 131 COVID-19 patients hospitalized at the University Hospital of Rome “Tor Vergata” and Bambino Gesù Children’s Hospital in Rome, during the period between March and May 2020. All patients were diagnosed with COVID-19 based on clinical evidence and confirmed by viral RNA detection at oropharyngeal and nasopharyngeal swabs by real-time PCR. The patients were clustered in three groups: (1) severe, according to respiratory impairment, requiring non- invasive ventilation; (2) extremely severe, defined as respiratory failure, requiring invasive ventilation and intensive care unit admission, (3) asymptomatic, according to absence of clinical symptoms and not requiring hospitalization or ventilation.

The majority of the enrolled patients were males (82 males, 49 females). Median age was 63.7 years (range: 2–92 years), seventy-eight patients were under 65 years old. Ten patients were children (median age was 11.5 years) showing a severe form of the disease but none of them had Kawasaki-like syndrome [12,14]. Peripheral blood samples were collected. The analytical procedure received approval by the local ethics committee at the University Hospital of Rome Tor Vergata (protocol no. 50/20). The study was conducted in agreement with the principles of the Declaration of Helsinki. Informed written consent was obtained from each patient. 

### 2.2. Whole Exome Sequencing and Data Preprocessing

Library preparation and whole exome capture were performed by using the Twist Human Core Exome Kit (Twist Bioscience, South San Francisco, CA, USA) according to the manufacture’s protocol and sequenced on the Illumina NovaSeq 6000 platform. The BaseSpace pipeline (Illumina, Inc., San Diego, CA, USA) and the TGex software (LifeMap Sciences, Inc., Alameda, CA, USA) were used for the variant calling and annotating variants, respectively. Sequencing data were aligned to the hg19 human reference genome. Based on the guidelines of the American College of Medical Genetics and Genomics, a minimum depth coverage of 30× was considered suitable for analysis. Variants were examined for coverage and Qscore (minimum threshold of 30) and visualized by the Integrative Genome Viewer (IGV).

### 2.3. Statistical Analysis

Differences in alleles frequencies between groups were evaluated by the Pearson χ^2^ test or by Fisher’s exact test, as requested according to the numbers of samples in the compared groups. *p*-values less than 0.05 were considered statistically significant. The Hardy–Weinberg equilibrium was evaluated, where possible, by the Pearson χ^2^ test.

## 3. Results

In total we identified in the four investigated genes, 17 genetic variants. The observed variants and their frequency distribution are reported in Table 1.

The variants frequencies detected in our examined cohort for the *BSG* and *DPP4* genes were not statistically different with respect to the frequencies listed in the GnomAD database [15] for the European non-Finnish population [16].

In *PCSK3* gene, we identified five different germline variants, one intronic (c.372+5G>A) and four missense (c.128C>T, p.Ala43Val; c.436G>A, p.Gly146Ser; c.893G>A, p.Arg298Gln; c.1906A>G, p.Ile636Val). The allelic frequency of these variants in GnomAD for the EUR reference population is very low and we detected a statistically significant difference for the c.893G>A, (p.Arg298Gln) compared to our cohort (*p* = 0.0047). Instead, the variant c.1906A>G, (p.Ile636Val) identified in a heterozygous male, is not present in the GnomAD database. (Table 1). The missense variants mapping on the protein known structure (G146S and R298Q) are both at more than 20Å from the protein active site.

In *TMPRSS2* gene, we identified five different missense variants (c.22G>C, p.Gly8Arg; c.23G>T, p.Gly8Val; c.193G>A, p.Ala65Thr; c.331G>A, p.Gly111Arg; c.589G>A, p.Val197Met). The last variant is the only one mapping on the catalytic domain, far from the active site (> 30Å). The missense variant c.193G>A, (p.Ala65Thr), and c.589G>A (p.Val197Met) were found in COVID-19 patients with frequencies in line with those reported for the reference population in the GnomAD database. Instead, a significant difference was detected for the c.331G>A, (p.Gly111Arg), a very rare allele in the European non-Finnish population of the GnomAD database, observed in our cohort in a heterozygous male patient (*p* = 0.0163). Furthermore, we observed a lower frequency of c.23G>T (p.Gly8Val) variant allele and of c.22G>C, p.Gly8Arg in COVID-19 patients, compared to allelic frequency of these variants in GnomAD for the EUR reference population (*p* = 0.0446 and *p* = 0.0228, respectively).

We also verified Hardy–Weinberg Equilibrium for those variants for which it was possible to calculate and the genotypes distributions resulted in HW equilibrium for all of them (Table 1). 

To predict the potential impact on proteins of the rare identified variants (two in the *PCSK3* gene and one in the *TMPRSS2* gene), we used different tools (PolyPhen2, Mutation Taster, SIFT) [17,18,19], and two ensemble score (MetaLR_pred, MetaSVM_pred.). For the variant c.893G>A (p.Arg298Gln) identified in the *PCSK3* gene, the in-silico analysis gave pathogenic computational verdict. The Arg residue at position 298 is a large hydrophilic amino acid, while the variant residue is a Gln, which has a neutral side chain. The sequence alignment of the furin protein with its orthologous proteins shows that the wild type residue is highly conserved in species ranging from human to zebrafish and lamprey (phastcons 46, in UCSC Genome browser) [20], implying an important functional or structural role for this residue in the furin protein. Also for the c.1906A>G (p.Ile636Val) in the *PCSK3* gene, the wild type residue is highly conserved in species ranging from human to zebrafish and lamprey, but the change Ile in Val may not lead to changes in function as both amino acid residues have an aliphatic side chain and are hydrophobic. Indeed, in silico analysis gave benign computational verdict. Concerning the variants, c.331G>A (p.Gly111Arg) in the *TMPRSS2* gene, the residue is conserved only in mammals and the in silico analysis gave benign computational verdict.

## 4. Discussion

We analyzed the genetic variants located on *TMPSSR2, PCSK3, DPP4,* and *BSG* coding-region in a representative cohort of Italian patients affected by COVID-19, in order to verify the hypothesis that the COVID-19 susceptibility is also influenced by genetic variability of genes coding for proteins involved in the entry of SARS-CoV-2 into target cells. By our preliminary study, we observed in the group of COVID-19 patients some differences in the frequencies of the genetic variants located in the two proteases studied, TMPRSS2 and PCSK3, compared to those reported in the GnomAD database.

Several studies have shown that SARS-CoV-2 entry is regulated by proteases. Proteolytic activation of S protein potentially leads to the final structural change of S2 needed for membrane fusion and the serine protease TMPRSS2 seems to be the main enzyme involved in this process [21]. In the COVID-19 patient group we identified the rare variant c.331G>A (p.Gly111Arg) localized in the *TMPRSS2* gene, with a higher frequency compared to the GnomAD database.

A study conducted on an Italian cohort has shown that Italians had a significant decrease in the burden of deleterious variants in *TMPRSS2* compared to the other Europeans populations, suggesting that they might have a higher level of TMPRSS2 protein or activity, which seems to represent a risk factor for a more severe disease course [22]. 

We also observed a lower frequency of c.23G>T (p.Gly8Val) and c.589G>A (p.Val197Met) polymorphic alleles in COVID-19 patients, compared to the allelic frequency of these variants in GnomAD for the EUR reference population. In particular, the missense variant c.589G>A is located in the extracellular SRCR (Scavenger Receptor Cysteine-Rich) domain of TMPRSS2, which interacts with external pathogens [23]. The SRCR domain guides the protease domain, correctly orienting it towards its substrate and when absent, the proteolytic activity of TMPRSS2 decreases considerably, as demonstrated by functional studies. [24]. The minor variant allele of this polymorphism could alter the processing of SARS-CoV-2 S protein and thus present a protective effect towards infection.

Moreover, from bioinformatics analyses conducted by Zerubin et al. [24], *TMPRSS2* was co-expressed with the main SARS-CoV-2 receptors (*ACE2* and *BSG*). In addition, the authors extrapolated nine drugs from the DRUGBANK database that could reduce the expression levels of *TMPRSS2*. In particular, two of these drugs (Paracetamol and Curcumin) are currently being administered to alleviate some symptoms in COVID-19 patients [25]. In a study in vitro using cell lines and primary pulmonary cells, an inhibitor of the protease activity of TMPRSS2, camostat mesylate, partially inhibited the entry of SARS-CoV-2 into these lung epithelial cells [5]. These data suggest that inhibition of this protein may be a potential intervention to be investigated.

TMPRSS2 was the first identified protease capable of cleaving the S protein, but recent studies highlighted that SARS-CoV-2 contains a canonical furin-cleavage site, lacking in the other SARS-like CoVs, which facilitates its entry into the cells [7]. Furin is a member of the evolutionarily ancient family of proprotein convertases, called PCSK. Humans encode nine members of this protease family (PCSK1–9), with PCSK3 representing furin [26]. *PCSK3* gene has low allelic frequencies of missense variants, as expected on the basis of GnomAD population data, but we have identified five of these rare variants in COVID-19 patients. In particular, the c.893G>A causes an amino acid change from arginine to glycine in a very highly conserved position (R298) near the substrate-binding residues (292-295). In silico analysis showed that the variant does not alter the protein structure, but it could influence the recognition of the target sequence RxxR within SARS-CoV-2 spike protein. 

Another interesting mutation identified in the *PCSK3* gene is c.1906A>G (p.Ile636Val). This variant was predicted benign by in silico tools and it was not found in the GnomAD database. Therefore, its functional meaning should be clarified in subsequent studies.

PCSK3 is regulated by a family of proteins containing the EMI domain, a cysteine-rich sequence of approximately 80 aminoacids [27]. The EMILIN/Multimerin family includes EMILIN1, EMILIN2, Multimerin1, and Multimerin2: in the future it could be interesting to study also their possible genetic variants. Indeed, the EMI domain is characterized by seven cysteine residues located at regular positions and this domain is located at the N-terminus in all the proteins in which it is expressed. EMILIN1, belonging to the EMILIN family, seems to bind upstream of the PCSK3 convertases and prevents the cleavage of its targets [28].

The choice of a patient group well clinically characterized and homogeneously and accurately diagnosed is an essential requirement in a genetic association study. In the case of COVID-19, the patients could have a high phenotypic variability, with different manifestations, ranging from very mild sub-clinical symptoms to ARDS. Therefore, it is difficult to define a homogeneous cohort, in which the patients under examination share symptoms and disease severity. Although the sample size appears sufficient to highlight differences in the genotypic frequency of the COVID-19 patient population compared to those reported in the GnomAD database, the number of subjects is too small to stratify them on the basis of clinical characteristics and clinical phenotypes. We consider this a limitation of our study. Besides, we consider that this preliminary study on the genetic variability of the genes involved in the entry of the virus could be useful for the scientific community. Moreover, an interesting data is that the three rare mutations whose frequencies were different with respect to the frequencies reported in the database, were all identified in the subgroup of “severe” patients. 

Although genetic variants in these genes may influence entry of the SARS-CoV-2, currently there are no functional studies that can demonstrate their active role in susceptibility to infection. We can only hypothesize that mutations in these genes could alter protease functionality and facilitate the entry of the SARS-CoV-2, providing further confirmation of the importance of the spike protein activation process. From the data that we obtained, there are several differences in some of the genetic variants distribution with respect to Gnomad frequencies, but very strong associations with the disease do not seem to emerge. It would therefore be interesting to investigate in the future genetic variants in the intronic or regulatory regions of these genes, for example in the 5 ‘regions upstream of the promoters or in the 3’ UTR sites, in the points of interaction with miRNAs. These preliminary results, although requiring further confirmation in larger independent cohorts and the population, as well as functional studies to evaluate the actual effect of the detected genetic variants, bring new supporting data to the role of human host genetic variability in the susceptibility to SARS-CoV-2 infection and its spread in human populations. 

## 5. Conclusions

We believe that the variability in host susceptibility to infections, including SARS-CoV-2, is due to an interplay of different factors and our contribution in the dissection of the genetic ones could give support to these kind of studies, that could also be useful in identifying a higher-risk population that could also be prioritized for vaccination.

## Figures and Tables

**Table 1 genes-11-01010-t001:** Comparison of allelic counts (variant vs. wild type alleles) between our Italian population of SARS-CoV2 Positive Patients and Europeans (GnomAD database).

Gene	Nr.	dbSNP	Position (Hg19)	Coding	Protein	Genotype (n)	Variant Type	Allelic Count	Allelic Count EUR (GnomAD)	Allelic Frequency	Allelic Frequency EUR (GnomAD)	*p*-Value	H.W.
*BSG*	1	rs201850688	Chr19: 572671	c.37C>G	p.Leu13Val	Het (1)	Missense_variant	1 vs. 261	87 vs. 66377	1/262 = 0.004	17240/86164 = 0.001	*p* = 0.2928	Not possible to calculate
	2	rs11551906	Chr19: 572680	c.46A>G	p.Thr16Ala	Het (2)	Missense_variant	2 vs. 260	1384 vs. 62996	2/262 = 0.008	1384/64380 = 0.021	*p* = 0.1919	In equilibrium
	3	rs144824657	Chr19: 577782	c.76G>T	p.Val26Phe	Het (1)	Missense_variant	1 vs. 261	414 vs. 4028078	1/262 = 0.004	414/40322 = 0.01	*p* = 0.5306	Not possible to calculate
	4	rs41276870	Chr19: 579501	c.417C>T		Het (1)	Splice_region_variant	1 vs. 261	1366 vs. 125692	1/262 = 0.004	1366/127058 = 0.01	*p* = 05376	Not possible to calculate
*DPP4*	1	rs116302758	Chr2: 162904013	c.95-2A>G		Het (6)	Splice_acceptor_variant	6 vs. 256	4933 vs. 122045	6/262 = 0.023	4933/126978 = 0.039	*p* = 0.2511	In equilibrium
	2	rs56179129	Chr2: 162890142	c.796G>A	p.Val266Ile	Het (2)	Missense_variant	2 vs. 260	688 vs. 126666	2/262 = 0.008	688/127354 = 0.005	*p* = 0.6557	In equilibrium
	3	rs115450134	Chr2: 162865748	c.1887+3G>A		Het (1)	Splice_region_variant	1 vs. 261	1069 vs. 127981	1/262 = 0.004	1069/129050 = 0.008	*p* = 0.7299	Not possible to calculate
*FURIN*	1	rs16944971	Chr15: 91419098	c.128C>T	p.Ala43Val	Het (1)	Missense_variant	1 vs. 261	273 vs. 106605	1/262 = 0.004	273/106878 = 0.0025	*p* = 0.4892	Not possible to calculate
	2	rs780909157	Chr15: 91419792	c.372+5G>A		Het (1)	Splice_region_variant	1 vs. 261	6 vs. 24650	1/262 = 0.004	6/24656 = 0.00024	*p* = 0.0713	Not possible to calculate
	3	rs201551785	Chr15: 91420189	c.436G>A	p.Gly146Ser	Het (1)	Missense_variant	1 vs. 261	51 vs. 129085	1/262 = 0.004	51/129136 = 0.00039	*p* = 0.1	Not possible to calculate
	4	rs769208985	Chr15: 91422046	c.893G>A	p.Arg298Gln	Het (1)	Missense_variant	1 vs. 261	1 vs. 111907	1/262 = 0.004	1/111908 = 0.0000089	***p* = 0.0047**	Not possible to calculate
	5	rs1236237792	Chr15: 91424629	c.1906A>G	p.Ile636Val	Het (1)	Missense_variant	1 vs. 261	Not reported	1/262 = 0.004	Not reported	-	Not possible to calculate
*TMPRSS2*	1	rs200291871	Chr21: 42879910	c.22G>C	p.Gly8Arg	Het (1)	Missense_variant	1 vs. 261	360 vs. 32572	1/262 = 0.004	360/32932 = 0.01	*p* = 0.5387	Not possible to calculate
	2	rs75603675	Chr21: 42879909	c.23G>T	p.Gly8Val	Het (55) Hom (20)	Missense_variant	95 vs. 157	14273 vs. 19331	95/262 =0.36	14273/33604 = 0.425	***p* = 0.0446**	In equilibrium
	3	rs61735791	Chr21: 42866439	c.193G>A	p.Ala65Thr	Het (2)	Missense_variant	2 vs. 260	365 vs. 128739	2/260 =0.008	365/129104 = 0.0028	*p* = 0.1708	In equilibrium
	4	rs114363287	Chr21: 42866301	c.331G>A	p.Gly111Arg	Het (1)	Missense_variant	1 vs. 261	7 vs. 127779	1/262 = 0.004	7/127786 = 0.000055	***p* = 0.0163**	Not possible to calculate
	5	rs12329760	Chr21: 42852497	c.589G>A	p.Val197Met	Het (33) Hom (6)	Missense_variant	45 vs. 217	29831 vs. 98773	45/262 =0.17	29831/128604 = 0.23	***p* = 0.0228**	In equilibrium

Significant differences are reported in bold; Het: Heterozygous; Hom: Homozygous, H.W.: Hardy–Weinberg.

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
