# Peer review of "COVID-19 and Genetic Variants of Protein Involved in the SARS-CoV-2 Entry into the Host Cells"

_genes, 2020, doi:10.3390/genes11091010_

Round 1

Reviewer 1 Report

 The authors studied 131 COVID-19 patients by exome sequencing of of the TMPRSS2, PCSK3, DPP4 and BSG genes. They identified 29 seventeen variants. Although genetic variants in these genes may influence entry of the SARS-CoV-2, the data didn't provide direct evidence for this, nor was there demonstration between these variants and host susceptibility and severity of the disease.  The authors may consider modify the title of the study and discuss from an angle that fit the data better. 

Author Response

REVIEWER 1

Q.: The authors studied 131 COVID-19 patients by exome sequencing of the TMPRSS2, PCSK3, DPP4 and BSG genes. They identified 29 seventeen variants. Although genetic variants in these genes may influence entry of the SARS-CoV-2, the data didn't provide direct evidence for this, nor was there demonstration between these variants and host susceptibility and severity of the disease. The authors may consider modify the title of the study and discuss from an angle that fit the data better.

Author response: The comment is appropriate. We modified the discussion, underlining the absence of direct functional evidence of the genetic variants identified. In fact, in the manuscript we only affirmed that we have observed differences in the frequencies of these genetic variants between COVID-19 patients and European populations reported in GnomAD database and we highlighted that ours are preliminary results to be confirmed in larger independent cohorts and populations. But we think that this preliminary information on the variability of the genes involved in virus entry can be useful to the scientific community

In addition, we have also modified the title of the manuscript as suggested

Reviewer 2 Report

General comments:

The authors have applied recent information about four proteins involved in the SARS-CoV-2 cell entry through proteolytic preprocessing of the spike protein. Identification of protein variants in humans may be relevant, for instance to predict susceptibility to disease development. Allelicity of the four genes was explored in a patient cohort of 131 patients and compared to the general trends in the European population. Despite only a few and subtle mutations were found, I think the strategy followed and the preliminary results may be of general scientific interest.

However, given the barely significant associations between their cohort and the global European population, authors should elaborate more the potential causality of the mutations in this concern and build some knowledge-based hypothesis. Protease versions associated to patients should show either higher catalytic activity or higher affinity to the viral protein than their non-associated counterparts. At the very least, authors may consider to explore the location of their significant reported positions respect to the active site of these proteins. Otherwise, the work is interesting but maybe too descriptive.

Minor comments:

_ L 117. Provide a reference for the GnomAD database.

_ L 159. In this sentence, it seems Italians are not Europeans. Please, rewrite.

_ L 190. The regulation hypothesis is of interest since mutations leading to enhanced activity/expression levels of these proteases may also be a pre-disposition factor. What is the mechanism of regulation of PCSK3 by the EMI domain proteins? This may offer a clue of what to search.

_ Please, state whether significant residue changes are conservative (same amino acid class) or not. This is also a way to estimate the severity of the mutations.

Author Response

REVIEWER 2

Q1.:General comments:

The authors have applied recent information about four proteins involved in the SARS-CoV-2 cell entry through proteolytic preprocessing of the spike protein. Identification of protein variants in humans may be relevant, for instance to predict susceptibility to disease development. Allelicity of the four genes was explored in a patient cohort of 131 patients and compared to the general trends in the European population. Despite only a few and subtle mutations were found, I think the strategy followed and the preliminary results may be of general scientific interest.

Thank you for the positive comments.

Q2: However, given the barely significant associations between their cohort and the global European population, authors should elaborate more the potential causality of the mutations in this concern and build some knowledge-based hypothesis. Protease versions associated to patients should show either higher catalytic activity or higher affinity to the viral protein than their non-associated counterparts. At the very least, authors may consider to explore the location of their significant reported positions respect to the active site of these proteins. Otherwise, the work is interesting but maybe too descriptive.

Author response: Thanks for the suggestion. We reported in the revised version the information about the distance between the variant residue and the protease active site, whenever possible. (see sentences in lines 125 and 129).

Q3: Minor comments:

_ L 117. Provide a reference for the GnomAD database.

Author response: We have added the reference in the manuscript, as suggested. Moreover, website reference (https://gnomad.broadinstitute.org/) is reported at the bottom of the text. (see sentence in line 119).

_ L 159. In this sentence, it seems Italians are not Europeans. Please, rewrite.

Author response: Thank you for the comments. The sentence has been modified.

_ L 190. The regulation hypothesis is of interest since mutations leading to enhanced activity/expression levels of these proteases may also be a pre-disposition factor. What is the mechanism of regulation of PCSK3 by the EMI domain proteins? This may offer a clue of what to search.

Author response: We improved the discussion, reporting the mechanism by which EMI proteins seems to modulate PCSK3 convertases (see sentence in line 204).

_ Please, state whether significant residue changes are conservative (same amino acid class) or not. This is also a way to estimate the severity of the mutations.

Author response: For the three rare identified variants with different frequencies compared to GnomAD database, we had predicted the potential impact on proteins with different tools (PolyPhen2, Mutation Taster, SIFT).  As suggested, we also evaluated significant residue changes are conservative or not and this information is now explicitly given. (see sentences in lines 144 and 149).

Round 2

Reviewer 1 Report

No further comment.

Reviewer 2 Report

The authors have addressed all my concerns.